# KEAP1, a cysteine-based sensor and a drug target for the prevention and treatment of chronic disease

Sharadha Dayalan Naidu[1] and Albena T. Dinkova-Kostova[1,2]

[1]Jacqui Wood Cancer Centre, Division of Cellular Medicine, School of Medicine, University of Dundee, Dundee, UK
[2]Department of Pharmacology and Molecular Sciences and Department of Medicine, Johns Hopkins University School of Medicine, Baltimore, MD, USA

ATD-K, 0000-0003-0316-9859

Review  

**Subject Area:**
biochemistry

**Keywords:**
KEAP1, NRF2, cysteine, anti-inflammatory, antioxidant, redox

**Author for correspondence:**
Albena T. Dinkova-Kostova
e-mail: a.dinkovakostova@dundee.ac.uk

Redox imbalance and persistent inflammation are the underlying causes of most chronic diseases. Mammalian cells have evolved elaborate mechanisms for restoring redox homeostasis and resolving acute inflammatory responses. One prominent mechanism is that of inducing the expression of antioxidant, anti-inflammatory and other cytoprotective proteins, while also suppressing the production of pro-inflammatory mediators, through the activation of transcription factor nuclear factor-erythroid 2 p45-related factor 2 (NRF2). At homeostatic conditions, NRF2 is a short-lived protein, which avidly binds to Kelch-like ECH-associated protein 1 (KEAP1). KEAP1 functions as (i) a substrate adaptor for a Cullin 3 (CUL3)-based E3 ubiquitin ligase that targets NRF2 for ubiquitination and proteasomal degradation, and (ii) a cysteine-based sensor for a myriad of physiological and pharmacological NRF2 activators. Here, we review the intricate molecular mechanisms by which KEAP1 senses electrophiles and oxidants. Chemical modification of specific cysteine sensors of KEAP1 results in loss of NRF2-repressor function and alterations in the expression of NRF2-target genes that encode large networks of diverse proteins, which collectively restore redox balance and resolve inflammation, thus ensuring a comprehensive cytoprotection. We focus on the cyclic cyanoenones, the most potent NRF2 activators, some of which are currently in clinical trials for various pathologies characterized by redox imbalance and inflammation.

## 1. Introduction

All living organisms are vulnerable to various chemical stressors derived from endogenous and exogenous sources, such as reactive oxygen species (ROS), reactive nitrogen species (RNS) and reactive lipid species (RLS), which play important roles in cell signalling, but when produced in excess lead to oxidative stress. Oxidation–reduction (redox) reactions are common in biology, and the maintenance of redox homeostasis is vital for the correct functioning of most biological processes [1]. Oxidative stress occurs when there is an excess of oxidants, and the antioxidants are insufficient for restoring the intracellular redox balance [2]. Some examples of sources of exogenous oxidative stressors are environmental pollutants, ultraviolet (UV) and ionizing radiation, and genotoxic agents. Endogenous stressors, usually produced intracellularly, are derived from metabolic processes such as mitochondrial respiration and inflammation. Exposure to these chemically reactive species promotes cellular macromolecular damage. Chronic oxidative stress has been implicated in the development and exacerbation of neurodegenerative diseases [3,4], cancer [5,6], diabetes [7,8], autoimmune [9], cutaneous [10–12], pulmonary [13,14] and cardiovascular [14,15] diseases, infection [16], inflammation [17], as well as aging [18–20]. Cells have evolved several mechanisms to combat these

**Figure 1.** (*a*) Domain structure of human NRF2. There are seven NRF2-ECH (Neh) domains found within NRF2. The N-terminal Neh2 domain contains the KEAP1 binding motifs DLG and ETGE. The Neh4 and 5 domains within the transcription factor are required for its transactivation and the proteins that have exhibited binding to this region are CREB (cAMP responsive element binding protein) binding protein (CBP), AXIN1, silencing mediator of retinoic acid and thyroid hormone receptor (SMRT1), Receptor-associated coactivator 3 (RAC3), nuclear matrix protein (NRP/B), casein kinase 2 (CK2), Brahma-related gene 1 (BRG1) and mediator complex subunit 16 (MED16). The Neh7 domain found in the middle of the NRF2 protein has been shown to interact with retinoid X receptor alpha (RXRα) as well as retinoic acid receptor alpha (RARα). Two motifs, DSGIS and DSAPGS are found within the Neh6 domain, and are important for the binding of β-TrCP to facilitate NRF2-degradation, where the binding is promoted upon glycogen synthase kinase β (GSK3-β)-mediated phosphorylation of the DSGIS motif. C-Jun N-Terminal Kinase (JNK) binds to the Neh6 domain and phosphorylates S335. The Neh1 domain comprises of the DNA-binding motif and the binding region for the sMAF proteins. The carboxy-terminal Neh3 domain is also important for transactivation of NRF2 and chromodomain helicase DNA binding protein 6 (CHD6) interacts with this domain. (*b*) Domain structure of human KEAP1. The KEAP1 protein is a substrate adaptor for the CUL3-based E3 ligase, and is sectioned into five domains: (1) N-terminal region (NTR); (2) Broad complex, tramtrack, and Bric à Brac (BTB) domain allows for the homodimerization of KEAP1 monomers as well as CUL3 binding; (3) Intervening region (IVR); (4) Kelch domain (KELCH) is a structure consisting of a six-bladed β-propeller, where one KELCH subunit within the KEAP1 homodimer binds to the DLG motif and the other binds to the ETGE motifs found within the Neh2 domain of NRF2; and (5) C-terminal region (CTR). The black vertical lines represent the positions of the 27 cysteine residues present within the protein. Cartoon and surface representations of the BTB (pale pink) (PDB ID: 4CXI), IVR (pale yellow) (modelled) and KELCH (pale blue) (PDB ID: 5WFV) were drawn with UCSF ChimeraX software using X-Crystallographic images deposited into the Protein Data Bank (rscb.org) or modelled using the web-based I-TASSER platform.

relentless chemical insults in order to reinstate the redox homeostasis. The activation of the KEAP1/NRF2/ARE pathway is one such mechanism, which orchestrates the upregulation of antioxidant, anti-inflammatory and other cytoprotective proteins.

# 2. The NRF2/KEAP1/ARE cytoprotective pathway

Under homeostatic conditions, the transcription factor nuclear factor erythroid-2 p45-related factor 2 (NRF2) (figure 1*a*), is continuously ubiquitinated and targeted for 26S proteasomal degradation by its negative regulator Kelch-like (ECH)-associated protein 1 (KEAP1) (figure 1*b*), which is a substrate adaptor for the Cullin 3 (CUL3)-ring box 1 (RBX1) E3-ubiquitin ligase system [21]. Electrophiles from endogenous and exogenous sources or other small molecules (termed inducers) which activate NRF2 are able to do so via inactivating KEAP1 by reacting with its cysteine(s)

residues or by disrupting the KEAP1:NRF2 protein–protein interaction (PPI) interface [22,23]. Consequently, KEAP1 is unable to target the transcription factor for degradation. The interactions between KEAP1 and NRF2 and the effect of inducers on NRF2 stabilization can be visualized by the imaging of live cells expressing KEAP1 and NRF2, each fused to a fluorescent protein [24,25]. Following KEAP1 inactivation, the newly synthesized or free NRF2 is able to accumulate and translocate into the nucleus where it heterodimerizes with a small musculoaponeurotic fibrosarcoma (sMAF) protein and binds to the antioxidant response elements (ARE) with the consensus sequence 5′-TGACxxxGC-3′ found in the promoters of its target genes [21]. The 605-amino acid long NRF2 protein belongs to the family of the Cap'n'Collar (CnC) basic leucine zipper (bZIP) transcription factors, and is composed of seven NRF2-ECH (Neh) domains which are highly conserved (figure 1*a*). The N-terminally lying Neh2 domain of NRF2 contains two KEAP1 (low- and high-affinity) binding motifs, which are the sequences DLG and ETGE, respectively [26].

## 2.1. NRF2-mediated antioxidant effects

Together, the NRF2 target genes (over 250) are involved in mounting a cellular defence response by encoding a large network of proteins, some of which catalyse phase I, II and III cytoprotective detoxification reactions, while others have antioxidant and anti-inflammatory properties [27]. NRF2 controls the cellular redox homeostasis by regulating key enzymes and proteins involved in processes such as the synthesis, utilization and regeneration of glutathione (GSH), thioredoxin (TXN), peroxiredoxin and NADPH production [28]. The activity of NRF2 is a major determining factor of the cellular redox state.

GSH is an essential thiol-based intracellular tripeptide that plays a vital role in the defence against cellular oxidative stress through its ability to neutralize ROS/RNS as well as electrophilic species [29]. Perturbed glutathione homeostasis has been implicated in numerous pathological conditions [30,31]. NRF2 regulates the gene expression of the catalytic subunit GCLC and the modifier subunit GCLM of γ-glutamate-cysteine ligase (GCL), the enzyme catalysing the rate-limiting step in the GSH biosynthesis [32], as well as the gene expression of the cystine/glutamate antiporter (SLC7A11, system xc−) [33] that is responsible for the import of cystine, which in turn is converted to cysteine, a GSH precursor. The flux of glutamine into anabolic pathways is enhanced under conditions of NRF2 activation [34] thus providing glutamate, the second GSH precursor; of note, glutamate is also necessary for the import of cystine by system xc−. The transporter SLC6A9, another NRF2-regulated gene, provides the third GSH precursor, glycine [35]. In addition to the biosynthesis of GSH, NRF2 also regulates the regeneration of GSH. The transcription factor controls the expression of glutathione peroxidase (GPX), which detoxifies peroxides to produce oxidized glutathione (GSSG). In turn, GSSG is a substrate for the NRF2-target glutathione reductase (GSR), which regenerates GSH from GSSG using NADPH as a hydride donor. Importantly, NRF2 is also involved in the regulation of cellular NADPH levels by controlling the gene expression of the four main enzymes involved in the generation of NADPH: isocitrate dehydrogenase 1 (IDH1), 6-phosphogluconate dehydrogenase (PGD), glucose-6-phosphate dehydrogenase (G6PD) and malic enzyme 1 (ME1) [34,36].

The role of NRF2 in the biosynthesis and maintenance of GSH is particularly important in the brain [37,38], and may also affect the metabolic glutamate–glutamine cycle that allows the inter-cellular exchange of these amino acids between neurons and astrocytes [39]. During neuronal development, expression of *NFE2L2* (the gene encoding NRF2) is repressed by promoter methylation [40], and NRF2 activity in astrocytes is critical for neuronal protection against oxidative stress [41]. In rapidly proliferating cells, such as cancer cells, NRF2 activation channels glucose through the pentose phosphate pathway [34], a major source of reducing equivalents for GSH regeneration, but also increases consumption of glutamate for GSH biosynthesis and glutamate secretion by system xc− [42].

## 2.2. NRF2-mediated anti-inflammatory effects

In addition to antioxidant, the activation of NRF2 has anti-inflammatory effects, which have been consistently observed in cellular and animal models, as well as in human intervention trials with pharmacological NRF2 activators. Thus, a recent analysis of peripheral blood mononuclear cells (PBMCs) isolated from human subjects following intervention with sulforaphane, a classical NRF2 activator, reported an increase in the expression of NRF2-target genes (i.e. NQO1, HO1, AKR1C1), which was accompanied by a decrease in inflammatory markers (i.e. IL-6, TNFα, IL-1β, COX2) [43]. NRF2 is critical for the resolution of inflammation. The endogenous mildly electrophilic anti-inflammatory mitochondrial immunometabolite itaconate, which accumulates to millimolar concentrations during the metabolic reprogramming in activated macrophages [44,45], is an NRF2 activator. In turn, NRF2 represses the expression of pro-inflammatory cytokines and the type I interferon (IFN) response, promoting the resolution of inflammation [45–49]. Interestingly, NRF2 is also important for the execution of inflammation. Itaconate is downregulated in dysfunctional macrophages from hypercholesterolemic mice, and the levels of NRF2 and the expression of its target genes are lower in lipopolysaccharide (LPS)-stimulated macrophages isolated from mice fed high-fat diet (HFD) compared to standard fat diet (SFD) [50]. These findings illustrate that systemic metabolic changes can suppress NRF2 and consequently interfere with metabolic reprogramming in immune cells, which is necessary for their effector functions.

Thus, the activation of NRF2 is an attractive therapeutic strategy to combat diseases characterized by chronic oxidative stress and inflammation as it provides a multi-targeted approach [51,52]. Indeed, in recent years the pharmaceutical industry has invested heavily in the development of pharmacological modulators of the KEAP1/NRF2/ARE pathway, and there are currently more than 15 ongoing clinical trials as well as a number of compounds undergoing preclinical testing for various disease indications [23,53].

# 3. KEAP1

KEAP1 is a highly conserved cysteine-rich 624-amino acid protein sharing approximately 92% sequence homology among the mammalian species (figure 2). The existence of KEAP1 was predicted in the 1980s, more than a decade before its discovery. Following a series of extensive structure-activity studies using quantitative chemical biology approaches, Paul Talalay and his associates observed that numerous structurally diverse inducers of the cytoprotective enzymes NAD(P)H:quinone oxidoreductase (NQO1) and glutathione *S*-transferases (GSTs) have a common chemical property, namely sulfhydryl reactivity. This seminal discovery rationalized the perplexing lack of structural similarity among inducers, and led to the explicit suggestion that the primary cellular sensor with which inducers react is a protein endowed with highly reactive cysteines [54,55]. The identification of KEAP1 as a negative regulator of NRF2 by Masayuki Yamamoto and his colleagues in 1999 [56] immediately turned attention to the cysteine residues of KEAP1. Human KEAP1 has 27 cysteines, whereas mouse KEAP1 has 25, nine of which (red boxes in figure 2) are flanked by basic amino acids. Cysteines are unique amino acids due to their sulfhydryl (thiol) functional group that performs various functions, including: (1) forming intra- and intermolecular covalent bonds with other cysteine thiols, (2) binding to metals and metalloids, and (3) undergoing reversible or irreversible oxidation upon reacting with oxidants [57]. The *pKa* value of cysteine (represented by the balance between the thiol and the thiolate anion) indicates its reactivity [58]. The lower the *pKa* value, where the formation of the thiolate anion

royalsocietypublishing.org/journal/rsob    Open Biol. **10**: 200105

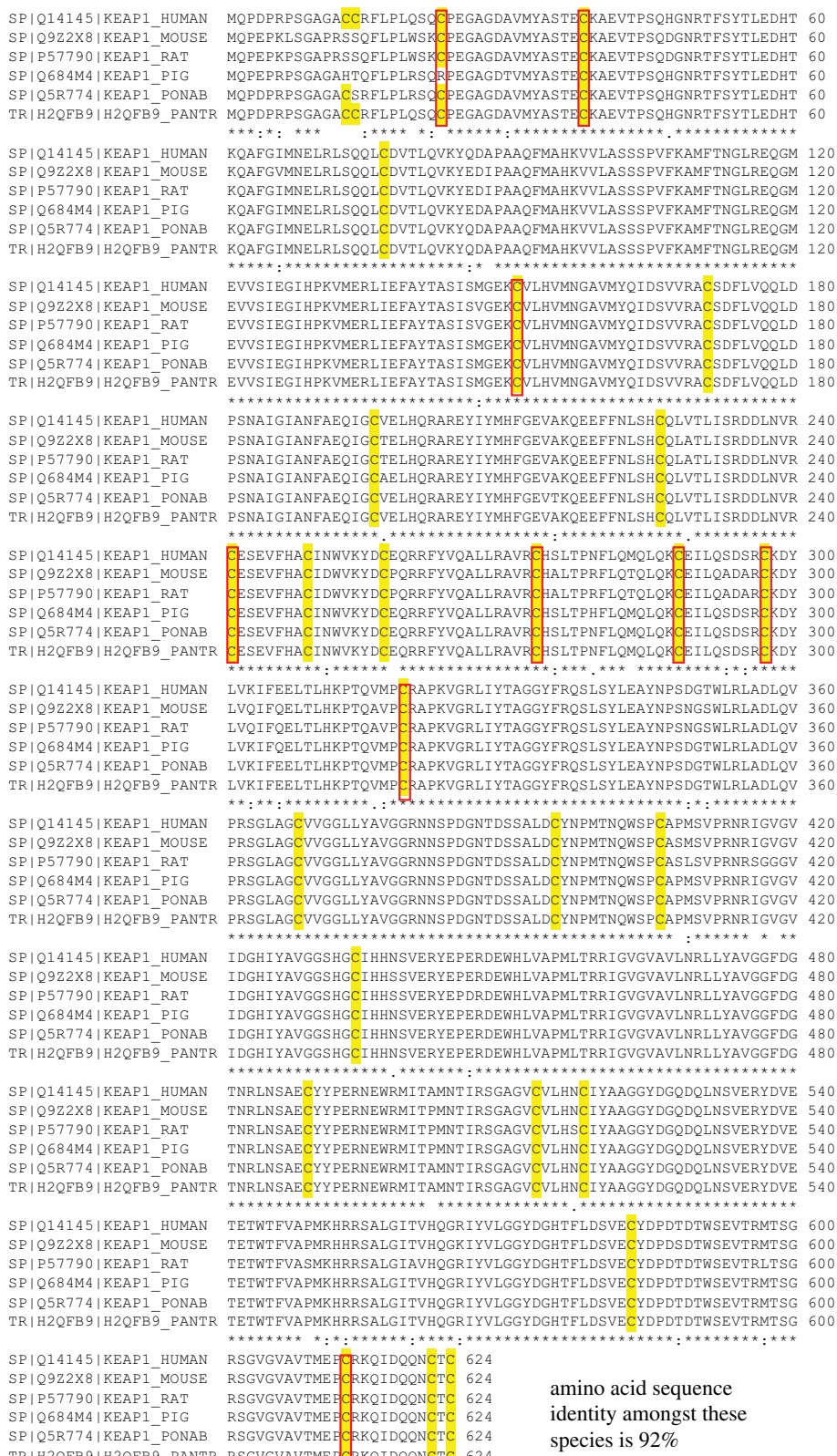

**Figure 2.** Amino acid sequence alignment of various mammalian KEAP1. Alignment performed using the web-based Clustal Omega program found within uniprot.org. The Uniprot IDs are listed. Q14145 (human), Q9Z2X8 (mouse), P57790 (rat), Q684M4 (pig), Q5R774 (orangutan), H2QFB9 (chimpanzee). The cysteine residues are highlighted in yellow and the cysteine residues which have neighbouring basic amino acids are boxed in red. The amino acid sequence identity among these species is 92%.

is favoured, the cysteine reactivity increases. The presence of basic amino acids in close proximity to a cysteine decreases the *pKa*, hence increasing its reactivity [59,60]. The identification of highly reactive cysteines within KEAP1 that serve as inducer sensors [61] solved the puzzling fact that many inducers are active at sub-micromolar concentrations despite the millimolar intracellular concentrations of glutathione.

## 3.1. KEAP1 structure and its cysteine sensors

KEAP1 is a homodimeric protein belonging to the BTB (Broad complex, Tramtrack, Bric-á-brac)-Kelch family of proteins, which are named Kelch-like 1 to 42 (KLHL1–42). All members of this family are able to bind to CUL3 through their BTB domain (figure 1*b*). The BTB domain is necessary for KEAP1

homodimerization, and it has been reported that mutation of S104 to an alanine residue prevents its homodimerization and causes NRF2 accumulation in the nucleus [62]. There are three cysteines present in the KEAP1 BTB domain, C77, C151 and C171. Single mutants of each of the cysteines in the BTB domain to a serine behave like the wild-type (WT) KEAP1 in terms of their ability to repress NRF2-mediated gene expression [63]. C151 is the most well characterized in the literature. Under basal conditions, the KEAP1 C151S mutant is able to mediate the degradation of NRF2, similarly to the WT protein [63–65]. Zhang et al. found that the BTB domain of KEAP1 protected KEAP1 from ubiquitin-mediated degradation [66]. Interestingly, it has been shown by several groups that under basal conditions, when subjected to SDS-polyacrylamide gel electrophoresis (PAGE), WT KEAP1 migrates as two distinct species, one at approximately 65 kDa and the other at 130 kDa and that the slower migrating species does not appear within the C151S mutant [63,67]. Zhang and colleagues suggested that the slow migrating species of KEAP1 is due to posttranslational modifications occurring on the protein, as the mutation of C151S prevents its occurrence. We have observed a similar effect following treatment of cells with the double Michael acceptor dibenzylidene acetone (DBA) (S.D.N. 2016, unpublished observations). Fourquet et al. have shown that the intensity of this slower migrating KEAP1 species, which is resistant to reducing agents, is increased upon exposure to oxidants and nitrosative agents. These authors subsequently exposed lysates from induced cells to the reducing agent β-mercaptoethanol and observed a complete reduction in the intensified slower migrating species of KEAP1 hence suggesting that this species mainly consists of the oxidized form of KEAP1 [67]. Most recently, C151 from one subunit of the KEAP1 dimer was shown to form a methylimidazole crosslink with R135 from the second subunit upon accumulation of the reactive metabolite methylglyoxal, the concentration of which is increased in the plasma of diabetic patients [68]. In all cases, these post-translational modifications of C151 result in dimerization of KEAP1, accumulation of NRF2 and activation of the NRF2-driven cytoprotective transcriptional program. Of note, methylglyoxal is a precursor of highly damaging advanced glycation end-products, and NRF2 regulates the expression of glyoxalase 1, the enzyme that detoxifies methylglyoxal, thus protecting against glycation [69].

The cysteine residues of KEAP1, for which chemical modifications by various electrophiles and oxidants have been either demonstrated directly or implicated based on mutagenesis analyses, are summarized in table 1. Some of the well-known NRF2 inducers that modify C151 are the isothiocyanate sulforaphane (SFN), the alkylating agent iodoacetamide (IAA), tert-butyl hydroquinone (tBHQ) and diethylmaleate (DEM) [64,65,73,92]. In 2010, using molecular modelling, McMahon et al. postulated that the reactivity of C151 in KEAP1 was due to the presence of five basic amino acid residues (H129, K131, R135, K150 and H154) located in close spatial proximity to C151 [64]. These five basic amino acids possess the ability to deprotonate the thiol group within C151, thereby, lowering its pKa. This results in the thiol group of C151 to exist as an anion under physiological pH conditions. Indeed, the authors showed that KEAP1 bearing the triple mutations K131M, R135M and K150M, lost the ability to sense electrophiles that specifically targeted C151 [64]. The crystal structure of the BTB domain of KEAP1 with the triterpenoid CDDO was solved in 2014 and deposited in the Protein Data Bank (rscb.org) with the accession number 4CXI [72]. We measured the distances of the 5 positively charged amino acids mentioned that were adjacent to C151 and found that R135 had the closest proximity to C151 with a distance of 3.6 angstroms (Å) (figure 3), which further supports the findings reported by McMahon and colleagues [64].

To date, there is no crystal structure of the intervening (IVR) domain of KEAP1 (aa 180 to 315) available. The KEAP1 IVR domain, flanked by the N-terminal BTB domain and the Kelch domain at the C-terminus, contains 8 cysteine residues (C196, C226, C241, C249, C257, C273, C288 and C297), of which, C273 and C288 are best characterized. Exposure to electrophiles or alkylating agents targeting the cysteines within the IVR domain, predominantly C273 and C288, leads to the inactivation of KEAP1 and subsequent activation of NRF2 [61,93,94]. Single or double mutations of C273 or C288 to serine or alanine render KEAP1 inactive with respect to its ability to repress and target for degradation NRF2 [63,64,93,95,96]. Since these C273S/A and C288S/A mutants inactivated KEAP1, they presented a difficulty to study the electrophiles that could potentially target these cysteines. By systematically mutating these cysteines, Saito and colleagues showed that single or double mutation of C273 and C288 to tryptophan or glutamic acid did not impede the KEAP1-mediated repression and degradation of NRF2 hence allowing to precisely identify electrophiles that are sensed by either or both of these cysteines [65]. In the report published by Saito et al. the authors showed that 15-deoxy-$\Delta^{12,14}$-prostaglandin J$_2$ (15d-PGJ$_2$) is sensed specifically by C288, extending the earlier observations by Levonen et al. who recognized the importance of cysteine thiols within KEAP1 for sensing electrophilic lipids [95]. We performed molecular modelling of the amino acid residues comprising the IVR domain using the web-based I-TASSER platform service [97–99], and found that it is comprised of nine α-helices (figure 4). The basic amino acids adjacent to C273 (i.e. R272 and H274) and C288 (i.e. K287) are expected to cause the deprotonation of the cysteine thiol groups, hence increasing their reactivity.

The Kelch domain of KEAP1 is evolutionarily conserved and contains nine cysteine residues at amino acid positions 319, 368, 395, 406, 434, 489, 513, 518 and 583. The first crystal structure of the KEAP1 Kelch domain was solved at a resolution of 1.85Å in 2004 by Li and colleagues [100]. Several crystal structures of both the human and murine KEAP1 Kelch domains with different resolutions and in combination with compounds or short peptide sequences of the Neh2 domain of NRF2 have since been reported [101–107]. The Kelch domain in KEAP1 contains six Kelch repeats that assemble into a six-bladed β-propeller structure (blades I-VI), where the C-terminal residues form the first strand in the first blade (figure 1b). Four-stranded antiparallel β-sheets form one blade, where the shortest β-sheet is found at the central core [100,108]. The Kelch domain also contains double glycine repeats (DGR), which are located at the terminal end of the β-sheets.

First discovered in the laboratory of Masayuki Yamamoto in 1999 by Itoh and colleagues, the N-terminal Neh2 domain within NRF2 has been since reported by various groups to bind to the KEAP1 Kelch domain [56]. Using nuclear magnetic resonance spectroscopy, Tong et al. discovered that the Neh2 domain was intrinsically disordered [26]. It was subsequently found that the evolutionarily conserved DLG [109] and ETGE [56] motifs within the NRF2-Neh2 domain

**Table 1.** Cysteine residues of KEAP1, for which chemical modifications by the indicated electrophiles and oxidants, have been implicated.

| | MCE-1 | MCE-23 | TBE-31 | RTA-408 | CDDO-Im | SF | PEITC | tBHQ | DEM | DMF | MEF | H₂O₂ | H₂S | NO | O⁻ NO₂ | 4- SNAP | 4- HNE | 15d- PGJ₂ | 8-NO cGMP | PGA₂ | As³⁺ | Cd²⁺ | Se⁴⁺ | Zn²⁺ | MeHg | Dex- Mes | 1,2- NQ | MIND4- 17 | (Z)- LIG | Ox- LIG | DATS | NAPQI | Acrolein | XH | 10- Shogaol | IAA | IAB | BMCC | ISO | Ebselen | 4- OI | GSSG |
|---|---|---|---|---|---|---|---|---|---|---|---|---|---|---|---|---|---|---|---|---|---|---|---|---|---|---|---|---|---|---|---|---|---|---|---|---|---|---|---|---|---|---|

3-ethynyl-3-methyl-6-oxocyclohexa-1,4-dienecarbonitrile (MCE-1) [70], 9a-ethynyl-3-oxo-9,9a-dihydro-3H-fluorene-2-carbonitrile (MCE-23) [70], (±)-(4bS,8aR,10aS)-10a-ethynyl-4b,8,8-trimethyl-3,7-dioxo-3,4b,7,8,8a,9,10,10a, octahydrophenanthrene-2,6-dicarbonitrile (TBE-31) [70], omaveloxolone (RTA-408) [71], 2-cyano-3,12-dioxooleana-1,9(11)-dien-28-oic acid (CDDO) [65,72], 2-cyano-3,12-dioxooleana-1,9(11)-dien-28-oic acid (CDDO-Im) [65,72], sulforaphane (SF) [63–65,73,74], phenethyl isothiocyanate (PEITC) [75], tert-butylhydroquinone (tBHQ) [75], 9a-ethynyl-hydroquinone (tBHQ) [63–65,73], diethylmaleate (DEM) [65,73], dimethyl fumarate (DMF) [73,76], monoethyl fumarate (MEF) [76], hydrogen peroxide (H₂O₂) [67,77,78], hydrogen sulfide (H₂S) [78], nitric oxide (NO) [64,67], nitro-oleic acid (OA-NO₂) [63,65], (±)-S-nitroso-N-acetylpenicillamine (SNAP) [65], 4-hydroxy-2-nonenal (4-HNE) [65], 15-deoxy-Δ12,14-prostaglandin J₂ (15d-PGJ₂) [65], prostaglandin A₂ (PGA₂) [79], 8-nitroguanosine 3′,5′-cyclic monophosphate (8-NO cGMP) [80], As³⁺ [64,65], Cd²⁺ [64], Zn²⁺ [64,65], Se⁴⁺ [64,65], methylmercury (MeHg) [82], dexamethasone 21-mesylate (Dex-Mes) [61], 1,2-naphthoquinone (1,2-NQ) [79,83], 5-nitro-2-[(15-phenoxymethyl)-4-phenyl-4H-1,2,4-triazol-3-yl(thio)pyridine (MIND4-17) [84], (Z)-ligustilide (Z-LIG) [85], oxidised-ligustilide (Ox-LIG) [85], diallyl trisulfide (DATS) [86], N-acetyl-p-benzoquinoneimine (NAPQI) [87], Se⁴⁺ [64], acrolein [64], xanthohumol (XH) [88], 1-(4-hydroxy-3-methoxyphenyl)-4-tetradecen-3-one (10-shogaol) [88], iodoacetamide (IAA) [87], N-biotinylhexylenediamine (IAB) [89], 1-biotinamido-4-(4′-[maleimidoethyl-cyclohexane]-carboxamido) butane (BMCC) [89], isoliquiritigenin (ISO) [88], 2-phenyl-1,2-benzisoselenazol-3(2H)-one (Ebselen) [79,90], 4-octyl itaconate (4-OI) [45] and glutathione disulfide (GSSG) [91].

royalsocietypublishing.org/journal/rsob Open Biol. 10: 200105

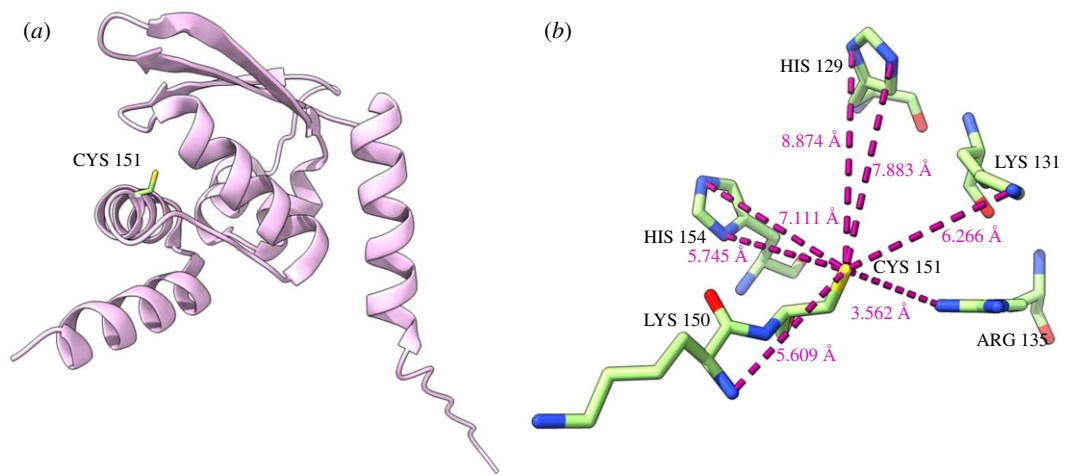

**Figure 3.** Structure of the human KEAP1 BTB domain. (*a*) Cartoon representation of the human KEAP1 BTB domain (pale pink) showing the side chain of CYS 151 in green. (*b*) The side chains of the basic amino acids (HIS 129, LYS 131, ARG 135, LYS 150 and HIS 154) adjacent to and surrounding the CYS 151 residues are represented with green stick drawings coloured by their elements. Structure drawn using UCSF ChimeraX software using the PDB accession 4CXI. The distances have been calculated in angstroms (Å) between these basic residues and CYS 151.

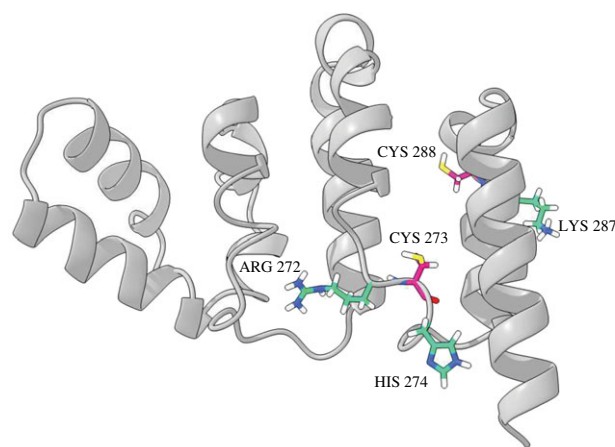

**Figure 4.** Modelled structure of the human KEAP1 IVR domain. Cartoon representation of the I-TASSER program modelled human KEAP1 IVR domain displaying 9 α-helices (grey). The basic amino acids (ARG 272 and HIS 274, green) found adjacent to the CYS 273 residue (pink) as well as the hydrophobic LYS 287 (green) residue found adjacent to CYS 288 (pink) where their side chains are represented with stick drawings coloured by their respective elements. The structure was drawn with UCSF ChimeraX software.

(figure 1*a*) were responsible for binding to the KEAP1 Kelch domains [26,110]. Compared with the ETGE motif, the DLG motif has a 200-fold lower affinity for the Kelch domain of KEAP1 [102]. The DLG and ETGE motifs flank a lysine-rich α-helix to allow for the conjugation of ubiquitin molecules by the activated ubiquitin conjugating E2 enzyme. Attachment of NRF2 to KEAP1 via both of these motifs is required for ubiquitination of the transcription factor, and a 'fixed-ends' or a 'hinge-and-latch' model for NRF2 ubiquitylation was proposed, where each binding motif of one molecule of NRF2 is tethered to a separate subunit of the KEAP1 homodimer [110,111]. Based on this knowledge, the Kelch domain of KEAP1 has become the target for the development of non-electrophilic NRF2 activators, which function as PPI inhibitors [112]. One example includes a series of 1,4-diphenyl-1,2,3-triazole compounds, which have been shown to disrupt the KEAP1:NRF2 PPIs *in vitro* using a fluorescence polarization assay, as well as in live cells expressing EGFP-

NRF2 and KEAP1-mCherry fusion proteins using a Förster resonance energy transfer-based system and multiphoton fluorescence lifetime imaging microscopy [113].

The C-terminal domain of KEAP1 is the home of three of the four cysteine sensors of KEAP1, which are used redundantly to mediate NRF2 activation in response to hydrogen peroxide ($H_2O_2$), the major ROS in redox regulation of biological processes. Very recently, Suzuki *et al.* [77] generated a construct for mammalian cell expression of a KEAP1 mutant, which lacks 11 out of the 25 cysteine residues of the murine protein. This mutant KEAP1 was still able to target NRF2 for ubiquitination and proteasomal degradation, but was unable to respond to most cysteine-reactive NRF2 activators, including $H_2O_2$. A series of elegant experiments involving mouse embryonic fibroblast cells expressing various KEAP1 cysteine mutants as well as five distinct KEAP1 mutant mouse lines revealed that KEAP1 uses C226, C613, C622 and C624 redundantly to sense $H_2O_2$ [77].

Although no crystal structure of full-length KEAP1 is available to date, a 24 Å resolution reconstituted electron microscopy (EM) structure has been described [114]. It shows that the KEAP1 dimer resembles a cherry-bob, where two large spheres, corresponding to the Kelch domains, are attached by short linker arms. Interestingly, each IVR domain surrounds the core of the Kelch domain, suggesting that chemical modifications of cysteines within the IVR domains may affect the KEAP1–NRF2 interactions through the Kelch domains.

## 3.2. KEAP1-CUL3 interaction

The primary system for protein degradation in the cell is the ubiquitin-proteasome system, where E3 ubiquitin ligases are essential components. Cullins are a family of hydrophobic proteins that confer substrate specificity by acting as scaffolds for E3 ubiquitin ligase complexes [115,116]. Thus far, in mammals, there have been seven Cullins (1, 2, 3, 4A, 4B, 5 and 7) identified. CUL3 is the only member of its family that is able to recognize BTB domains-containing proteins [116]. Since KEAP1 contains BTB domains, in 2004, Kobayashi and colleagues hypothesized that under basal conditions, CUL3 could be mediating the degradation of NRF2 through binding of the substrate adaptor KEAP1 [93]. Indeed, three independent groups simultaneously

**Figure 5.** Chemical structures of selected cyclic cyanoenone NRF2 activators.

discovered that KEAP1 forms a functional E3 ubiquitin ligase complex with CUL3/RBX1 [93,117,118]. Shortly after, Furukawa & Xiong reported that KEAP1 and CUL3 binding occurs between the BTB-domain of the former and the N-terminal domain of the latter [119]. Subsequently, it was discovered that CUL3 homodimerization requires the presence of its N-terminal domain and is dependent on its interaction with BTB domain-containing substrates (e.g. KEAP1) which also are able to homodimerize at their BTB domains [120].

# 4. The cyclic cyanoenones, the most potent class of NRF2 activators

To date, the cyanoenone triterpenoids are the most potent NRF2 activators known. They were designed and developed starting from the natural product oleanolic acid [121–123], and new generations of analogues have been synthesized [124]. These semi-synthetic compounds are highly electrophilic, bind covalently and reversibly to sulfhydryl groups [125], and have favourable pharmacokinetic and pharmacodynamic profiles *in vivo*, including in humans [126–128]. Currently, two cyanoenone triterpenoids are in clinical trials led by Reata Pharmaceuticals (USA) and Kyowa Hakko Kirin (Japan) for the treatment of diseases, which have been linked with chronic inflammation and abnormal redox homeostasis. One is CDDO-Me (2-cyano-3,12-dioxooleana-1,9(11)-dien-28-oic acid methyl ester; bardoxolone methyl, figure 5a) for the treatment of connective tissue disease–pulmonary arterial hypertension, pulmonary hypertension, Alport's syndrome, polycystic kidney disease, renal insufficiency and liver disease. The other is RTA-408 (omaveloxolone, figure 5b) for the treatment of Friedreich's ataxia, mitochondrial myopathy, ocular inflammation, ocular pain, corneal endothelial cell loss, cataract surgery, melanoma and radiation dermatitis in breast cancer patients [23]. Recently, Reata Pharmaceuticals reported the evaluation of the pharmacokinetics and tissue distribution of orally administered RTA-408 to cynomolgus monkeys after single and multiple oral doses, and the initial results from a

clinical trial in Friedreich's ataxia patients [126]. Dose-dependent plasma levels of RTA-408 and induction of NRF2 target genes were detected in peripheral blood mononuclear cells, liver, lung, and brain of the animals. In patients, improvements in neurological functions were observed at doses of 80 mg or greater; these doses resulted in plasma drug concentrations consistent with those inducing NRF2 target genes in animals.

To improve their potencies as anti-inflammatory agents and understand the details of their mechanism of action, numerous pentacyclic, tricyclic and monocyclic compounds containing cyanoenone functionalities were designed, synthesized and tested for their anti-inflammatory and NRF2-inducing activities in a programme of work led by Michael Sporn, Gordon Gribble, Tadashi Honda and Karen Liby at Dartmouth College [121,129–131]. In collaboration with the laboratory of Paul Talalay, these researchers found a linear correlation ranging over six orders of magnitude of concentrations between the potencies of 18 pentacyclic derivatives to inhibit inducible nitric oxide synthase (iNOS) and to activate the prototypic NRF2 target enzyme NQO1 [132]. Subsequently, this correlation was confirmed more broadly, for all main classes of NRF2 activators [133]. The high potency of the triterpenoid analogues in inducing NRF2 and inhibiting inflammation requires the presence of activated Michael reaction (enone) functions at critical positions in rings A and/or C. Among the cyclic cyanoenone derivatives, the acetylenic tricyclic bis(cyanoenone) TBE-31 is an exceptionally potent inducer (figure 5c). TBE-31 is active at sub- to low-nanomolar concentrations, with Concentration that Doubles the specific enzyme activity of NQO1 in murine Hepa1c1c7 cells (CD value) of 0.9 nM [134–137]. This compound is highly bioavailable and suitable for chronic oral administration [138,139]. The presence of two cyanoenone functionalities (in rings A and C) integrated in a three-ring structure confers particularly high inducer potency [138]. Another tricyclic cyanoenone, MCE-23 contains an identical ring C present in TBE-31, however, it does not have the cyanoenone moiety in its ring A (figure 5d), and is comparatively less potent, with a CD value of 41 nM [140]. MCE-1, a monocyclic cyanoenone (figure 5e), contains the

ring C of MCE-23 and TBE-31. Similar to MCE-23 and TBE-31, MCE-1 also induces NQO1 (CD = 22 nM) [131,138]. All of these cyanoenones exhibit anti-inflammatory activity in RAW 264.7 cells as well as primary macrophage (PM$\Phi$) cells derived from mice [131,138]. Furthermore, in a murine inflammation-mediated depression model, MCE-1 and TBE-31 exhibit anti-depressant effects [141].

CDDO-Me, RTA-408, MCE-1, MCE-23 and TBE-31 possess electrophilic Michael acceptor(s) within their chemical structures, and are therefore extremely reactive with sulfhydryl groups. Early studies employing ultraviolet–visible (UV-VIS) spectroscopy had shown that compounds of this class react with cysteines in KEAP1, but the identity of the specific cysteine sensor(s) within the protein was not known [132,136]. In collaboration with Takafumi Suzuki and Masayuki Yamamoto (Tohoku University), we generated KEAP1-knockout mouse embryonic fibroblast (MEF) cells that were reinstated with wild-type or various cysteine mutants of KEAP1, and monitored the stabilization of NRF2 upon exposure to cyanoenones [70]. In addition, we isolated PM$\Phi$ cells from wild-type KEAP1 (KEAP$^{+/+}$) or the knock-in mutant KEAP-C151S (KEAP$^{C151S/C151S}$) mice, which were generated by use of the CRISPR/Cas9 technology. This study revealed that C151 is the primary sensor for the cyanoenone class of NRF2 inducers, irrespective of their molecular shape or size. Furthermore, C151S mutation in KEAP1 (i.e. in the KEAP1$^{C151S/C151S}$ PM$\Phi$ cells) not only abolished the inducer activity of low concentrations of TBE-31, but it also diminished its anti-inflammatory activity. This effect was confirmed using transgenic mice expressing human interleukin 6 (IL-6)-luciferase reporter that were either KEAP1 wild-type or KEAP$^{C151S/C151S}$ mutant. Taken together, these experiments highlight the anti-inflammatory effect of NRF2 activation.

It is noteworthy that, although C151 is the primary sensor for the cyanoenone class of NRF2 inducers, the concentration of the inducer is critical for on-target selectivity. Thus, at low cyanoenone concentrations, C151 is essential for NRF2 stabilization, however, at higher cyanoenone concentrations, NRF2 stabilization proceeds in the absence of C151 [70]. These findings underscore the importance of the inherent flexibility of the sensor cysteines in KEAP1, explain the apparent discrepancies between the results from some of the published studies, and highlight the immense importance of determining the accurate dose of even the most selective electrophilic NRF2 inducer for achieving on-target selectivity.

Because RTA-408 can cross the blood–brain barrier and is currently in clinical development [23], its disease-modifying efficacy was tested in a rat model of status epilepticus, a disease, where cytotoxicity and inflammation constitute major pathogenic drivers. In a study led by Matthew Walker, Andrey Abramov and their colleagues at University College London, it was found that 3 daily doses of RTA-408 given in the first week of established disease potently inhibited epileptogenesis during the subsequent 12 weeks, and preserved both neurons and astrocytes in the hippocampus of the animals [71]. This remarkable effect indicates that breaking the vicious circle of redox imbalance causing macromolecular damage and cell death triggering inflammation leading to neuronal death and seizures, which in turn cause more neuronal death and greater inflammation, and more seizures, can be highly effective in managing this disease. Most importantly, the unprecedented high efficacy of RTA-408 in this model, and the fact that it is currently in clinical trials, suggests the potential for this drug as a disease-modifying treatment in epilepsy and perhaps other neurological conditions.

## 5. Concluding remarks

The extraordinary ability of KEAP1 to sense a multitude of inducers that vary in shape, size and reactivity, and are able to 'read' an intricately complex 'cysteine code', coupled with its intrinsic flexibility, guarantees a finely tuned, and tightly regulated antioxidant response. More than two decades of research conducted by numerous independent groups of investigators has convincingly demonstrated that the ability to mount this cytoprotective response is critical for adaptation and survival, and can be exploited to protect against or delay the onset of pathological processes, particularly those that involve oxidative stress and inflammation. Indeed, KEAP1 is the target of several small-molecule NRF2 activators, which are currently in clinical trials and hold promise for the prevention and treatment of chronic disease.

Data accessibility. This article has no additional data.

Competing interests. A.T.D-K. is a member of the Scientific Advisory Board of Evgen Pharma and a consultant for Aclipse Therapeutics and Vividion Therapeutics.

Funding. We thank Cancer Research UK (C20953/A18644) and Reata Pharmaceuticals for financial support.

Acknowledgements. We are immensely grateful to Michael Sporn, Gordon Gribble, Tadashi Honda and Karen Liby (Dartmouth College, USA) for introducing us to the cyanoenone class of NRF2 activators and most enjoyable collaborative interactions.

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
