## [Reviewer comments · Open Biology]

Review History

RSOB-20-0105.R0 (Original submission)

Review form: Reviewer 1

Recommendation

Accept with minor revision (please list in comments)

Do you have any ethical concerns with this paper?

No

Comments to the Author

This is a very updated and comprehensive study about the structure of KEAP1 and its regulation by sulfhydryl modification. The study also provides relevant information about current multiple electrophiles and the triterpenoids under clinical study. I recommend publication, but considering the applicability of this study to medicine and industry, I would suggest that the authors include a short comment and reference to paper Robledinos-Anton et al OMLC, 2019, in which clinical trials and patents about NRF2-targeting compounds are summarized. Although it is not essential for this review, the authors might consider adding at the end of point 2.1. a paragraph explaining the metabolic crosstalk between astrocytes and neurons regarding Glu/Gln cycle and GSH cycle. This would help stressing the relevance of KEAP1 regulation in a specific context.

Review form: Reviewer 2

Recommendation

Accept with minor revision (please list in comments)

Do you have any ethical concerns with this paper?

No

Comments to the Author

The manuscript by Naidu and Dinkova-Kostova is focused on the critical review of the KEAP1 as a drug target in the chronic diseases. Nrf2 and KEAP1 are one of the most promising modern drug target with growing interest to this field. The authors summarised view on the mechanisms of activation of Nrf2 and described current drugs targeted to KEAP1. This is interesting and important manuscript and I have only minor comment.

1. Description of the Table 1 (page 12) is confusing and could be better coordinated with table.

Decision letter (RSOB-20-0105.R0)

18-May-2020

Dear Professor Dinkova-Kostova,

We are pleased to inform you that your manuscript RSOB-20-0105 entitled "KEAP1, a cysteine-based sensor and a drug target for the prevention and treatment of chronic disease" has been accepted by the Editor for publication in Open Biology. The reviewer(s) have recommended publication, but also suggest some minor revisions to your manuscript. Therefore, we invite you to respond to the reviewer(s)' comments and revise your manuscript.

Please submit the revised version of your manuscript within 7 days. If you do not think you will be able to meet this date please let us know and we can extend this deadline for you.

1) A text file of the manuscript (doc, txt, rtf or tex), including the references, tables (including captions) and figure captions. Please remove any tracked changes from the text before submission. PDF files are not an accepted format for the "Main Document".

2) A separate electronic file of each figure (tiff, EPS or print-quality PDF preferred). The format should be produced directly from original creation package, or original software format. Please note that PowerPoint files are not accepted.

3) Electronic supplementary material: this should be contained in a separate file from the main text and meet our ESM criteria (see <http://royalsocietypublishing.org/instructions-authors#question5>). All supplementary materials accompanying an accepted article will be treated as in their final form. They will be published alongside the paper on the journal website and posted on the online figshare repository. Files on figshare will be made available approximately one week before the accompanying article so that the supplementary material can be attributed a unique DOI.

Online supplementary material will also carry the title and description provided during submission, so please ensure these are accurate and informative. Note that the Royal Society will not edit or typeset supplementary material and it will be hosted as provided. Please ensure that the supplementary material includes the paper details (authors, title, journal name, article DOI). Your article DOI will be 10.1098/rsob.2016[last 4 digits of e.g. 10.1098/rsob.20160049].

4) A media summary: a short non-technical summary (up to 100 words) of the key findings/importance of your manuscript. Please try to write in simple English, avoid jargon, explain the importance of the topic, outline the main implications and describe why this topic is newsworthy.

Images

Data-Sharing

It is a condition of publication that data supporting your paper are made available. Data should be made available either in the electronic supplementary material or through an appropriate repository. Details of how to access data should be included in your paper. Please see <http://royalsocietypublishing.org/site/authors/policy.xhtml#question6> for more details.

Data accessibility section

Sincerely,

The Open Biology Team

<mailto:openbiology@royalsociety.org>

Reviewer(s)' Comments to Author:

Referee: 1

Comments to the Author(s)

This is a very updated and comprehensive study about the structure of KEAP1 and its regulation by sulfhydryl modification. The study also provides relevant information about current multiple electrophiles and the triterpenoids under clinical study. I recommend publication, but considering the applicability of this study to medicine and industry, I would suggest that the authors include a short comment and reference to paper Robledinos-Anton et al OMLC, 2019, in which clinical trials and patents about NRF2-targeting compounds are summarized. Although it is not essential for this review, the authors might consider adding at the end of point 2.1. a paragraph explaining the metabolic crosstalk between astrocytes and neurons regarding Glu/Gln cycle and GSH cycle. This would help stressing the relevance of KEAP1 regulation in a specific context.

Referee: 2

Comments to the Author(s)

The manuscript by Naidu and Dinkova-Kostova is focused on the critical review of the KEAP1 as a drug target in the chronic diseases. Nrf2 and KEAP1 are one of the most promising modern drug target with growing interest to this field. The authors summarised view on the mechanisms of activation of Nrf2 and described current drugs targeted to KEAP1. This is interesting and important manuscript and I have only minor comment.

1. Description of the Table 1 (page 12) is confusing and could be better coordinated with table.

Author's Response to Decision Letter for (RSOB-20-0105.R0)

See Appendix A.

Decision letter (RSOB-20-0105.R1)

22-May-2020

Dear Professor Dinkova-Kostova

We are pleased to inform you that your manuscript entitled "KEAP1, a cysteine-based sensor and a drug target for the prevention and treatment of chronic disease" has been accepted by the Editor for publication in Open Biology.

Sincerely,
The Open Biology Team
mailto:openbiology@royalsociety.org

Appendix A

We would like to thank the Editor and the two reviewers for the positive comments and constructive criticism on our review article '**KEAP1, a cysteine-based sensor and a drug target for the prevention and treatment of chronic disease**', and for the opportunity to submit a revised version.

Below are our point-to point-responses. The changes from the originally submitted manuscript are shown in blue color in the revised version.

Referee: 1

"I recommend publication, but considering the applicability of this study to medicine and industry, I would suggest that the authors include a short comment and reference to paper Robledinos-Anton et al OMLC, 2019, in which clinical trials and patents about NRF2-targeting compounds are summarized."

We agree, and have included this reference at the end of the first paragraph on the top of page 4, reference 53.

"Although it is not essential for this review, the authors might consider adding at the end of point 2.1. a paragraph explaining the metabolic crosstalk between astrocytes and neurons regarding Glu/Gln cycle and GSH cycle. This would help stressing the relevance of KEAP1 regulation in a specific context."

We thank the reviewer for this excellent suggestion, and have included a new paragraph at the end of section 2.1 on page 3.

Referee: 2

"Description of the Table 1 (page 12) is confusing and could be better coordinated with table."

We thank the reviewer for pointing this out. We have now changed the title of the Table to improve clarity, and listed the names of the compounds and abbreviations in a footnote. We also slightly changed the way by which we referred to the table in the text (the beginning of the second paragraph on page 5).

In addition to these changes recommended by the two reviewers, on page 7, third paragraph, we added a new paragraph referring to the electron microscopy (EM) structure of KEAP1.

Once again, we thank the Editor and the two reviewers for their constructive criticism. We hope that the changes have improved the manuscript, and it is now acceptable for publication.